

# Migraine eye: correlation between migraine and the retina

Lunla Udomwech[1], Rini Sulastiwaty[2] and Doungkamol Siriarchawawat[3]

[1] Department of Ophthalmology, School of Medicine, Walailak University, Thasala, Nakhonsithammarat, Thailand
[2] Glaucoma Service, Jakarta Eye Center, Jakarta, Indonesia
[3] Division of Neurology, Department of Internal Medicine, School of Medicine, Walailak University, Thasala, Nakhonsithammarat, Thailand

## ABSTRACT

**Background:** Activation of the trigeminal vascular system in migraine releases vasoactive neurotransmitters, causing abnormal vasoconstriction, which may affect the ocular system, leading to retinal damage. The purpose of our study was to determine whether there are differences in each retinal layer between migraine patients and healthy subjects.

**Methods:** A case-control study recruited 38 migraine patients and 38 age- and sex-matched controls. Optical coherence tomography was used to measure the thickness of the peripapillary and macular retinal nerve fiber layer (pRNFL and mRNFL), ganglion cell layer (GCL), inner plexiform layer (IPL), and inner nuclear layer (INL).

**Results:** The mean ages of the migraine patients and controls were 36.29 ± 9.45 and 36.45 ± 9.27 years, respectively. Thirty-four patients (89.48%) in both groups were female. The mean disability score was 19.63 ± 20.44 (indicating severe disability). The superior-outer INL of migraine patients were thicker than controls. Thickness of the GCL at temporal-outer sector and mRNFL at the superior-outer sector of the headache-side eyes was reduced. However, the INL of the headache-side-eye showed negative correlation with the disability score. This is the first study having found thinning of the GCL and mRNFL of the headache-side eyes. The INL was also thickened in migraines but showed negative correlation with the disability score.

**Conclusions:** Increased INL thickness in migraine patients may result from inflammation. The more severe cases with a high disability score might suffered progressive retinal neuronal loss, resulting in thinner INL than less severe cases.

# INTRODUCTION

Migraine is a common primary headache disorder that affects patients worldwide. The prevalence among the Thai population is 29.1%, which is higher than that in the United States (11.7%) (*Phanthumchinda & Sithi-Amorn, 1989*). Migraine is characterized by recurrent attacks of throbbing, unilateral, and usually severe headaches (*Headache Classification Committee of the International Headache Society, 2018*). The understanding

Corresponding author
Doungkamol Siriarchawawat,
doungkamol.siri@gmail.com

of migraine pathophysiology has drastically improved in the last two decades. It is generally accepted that migraine attacks occur through the activation and sensitization of the trigeminal vascular system (TGVS) (*Russo, Tessitore & Tedeschi, 2013*). This activation releases pain-producing vasoactive neurotransmitters, causing abnormal vasoconstriction and aura symptoms when this phenomenon occurs in the occipital lobe.

Using optical coherence tomography (OCT), several studies have shown that vascular changes also occur in the ocular system during migraine attacks (*Killer, Forrer & Flammer, 2003*; *Abdul-Rahman, Gilhotra & Selva, 2011*). Vasoconstriction can cause damage to the retinal nerve fiber layer (RNFL) and retinal ganglion cell layer (GCL) (*Lin et al., 2021*), detectable by the thinning of these layers from OCT, although the results of such studies are discrepant. The purpose of our study was to determine whether there are differences in retinal thickness between migraine patients and healthy subjects. In migraine group, we evaluate RNFL thickness on the side of the headache (headache-side eye) compared to contralateral side (non-headache-side eye) and evaluate the thicknesses of the nasal and temporal sides of each eye in respect to the optic pathway. We measured the thickness of the peripapillary retinal nerve fiber layer (pRNFL), macular retinal nerve fiber layer (mRNFL), ganglion cell layer (GCL), inner plexiform layer (IPL), and inner nuclear layer (INL) using OCT to assess the correlation between these structural changes and migraine parameters, including disease duration, attack frequency per month, and disability score (Thai-MIDAS; the Thai version of Migraine Disability Assessment score).

## MATERIALS AND METHODS

**Participants:** this cross-sectional descriptive study was conducted at Walailak University Hospital (Nakhon Si Thammarat, Thailand) between July 2021 and August 2022. This prospective study was approved by the Walailak Ethics Committee (WUEC-21-166-01). Written informed consent was obtained from all participants after a full explanation of the study. This study complied with the principles of the Declaration of Helsinki and the International Conference on Harmonization of Good Clinical Practice.

We recruited migraine patients with and without aura who were diagnosed by a neurologist at the neurology outpatient clinic. All patients met the migraine diagnosis according to the criteria of the International Classification of Headache Disorders (ICHD-3 beta) (*Headache Classification Committee of the International Headache Society (IHS), 2013*). Migraine disability scores were assessed by a neurologist using a migraine disability questionnaire: Migraine Disability Assessment Score Thai Version (Thai-MIDAS), developed by *Vongvaivanich et al. (2018)*. Patients were interviewed about the type of migraine (with or without aura), the pattern of aura, duration of migraine history, the number of attacks per month (in the previous three months), antimigraine medication, impact of headaches on daily life, and pain severity scores. The Thai-MIDAS scores were divided into four grades: grade I, infrequent disability; grade II, mild disability; grade III, moderate disability; and grade IV, severe disability. We excluded patients with previous retinal disorders, glaucoma, previous ocular surgery/laser, and disorders of the cornea or lens that prevented the ophthalmologist from evaluating retinal thickness precisely.

The control group consisted of age- and sex-matched healthy controls (HCs) who were chosen prospectively and reported either fewer than three headaches in the past year or headaches not meeting the migraine ICHD-3 beta criteria. The exclusion criteria were a history of central nervous system disorders and ocular disorders as mentioned above.

**Ophthalmic evaluation and OCT:** all participants underwent a full ophthalmic examination by an ophthalmologist, including best-corrected visual acuity (BCVA) measured by the Early Treatment Diabetic Retinopathy Study (ETDRS) chart, intraocular pressure (IOP) measured by the iCare tonometer (Finland Oy, Vantaa, Finland), pupil size and reactivity, slit lamp examination, and fundus examination by BQ-900 slit lamp biomicroscope (Haag-Streit, Berne, Switzerland) at the ophthalmology clinic of Walailak University Hospital.

OCT was performed without pupillary dilation with the SPECTRALIS OCT2 Module (Heidelberg, Heidelberg, Germany). The image quality was assessed. The measurement of peripapillary RNFL included six sextants from each eye: temporal, nasal, temporal superior, nasal superior, temporal inferior, and nasal inferior. The GCL thickness, macular RNFL thickness, IPL thickness, and INL thickness were divided into eight regions according to ETDRS: temporal-inner, superior-inner, nasal-inner, inferior inner, temporal-outer, superior-outer, nasal-outer, and inferior-outer.

Both eyes of each participant were included for comparison of parameters between the migraine and healthy control groups. The sample size calculation was based on a study by *Tunç et al. (2017)* using the two independent means equation, with an alpha of 0.05 and a power of 80%. The predicted sample size was 40 patients in both the migraine and healthy control groups (a total of 80 participants).

Statistical analysis was performed using SPSS software version 18 (SPSS Inc., Chicago, IL, USA). Continuous data are presented as means and standard deviations (SDs); categorical data are presented as frequencies and percentages. The paired t-test was used to compare the pRNFL, mRNFL, GCL, IPL, and INL thicknesses in each group, and independent t-tests for means of variables from different groups. The difference is expressed as a probability value (*p*-value) and was considered significant at <0.05. ANOVA and *post-hoc* analysis among control eyes, headache-side eyes, and non-headache-side eyes are performed. The correlation coefficients (r) from Pearson's correlation test were analyzed to assess the correlation between ThaiMIDAS scores, the frequency of attacks, and the thickness of all areas. The 95% confidence interval (95% CI) was computed using a bootstrap approach to correct for multiple comparisons in the correlation analysis.

## RESULTS

We recruited 40 migraine patients and 40 healthy individuals for this study; two from each group were excluded due to poor image quality of the OCT, leaving 38 subjects in each group. Table 1 summarizes the demographic and clinical characteristics of the participants. The mean ages of the migraine patients and controls were 36.29 ± 9.45 and 36.45 ± 9.27 years, respectively. Thirty-four patients (89.48%) in both groups were female. The mean duration of migraine was 7.18 ± 6.05 years. The mean frequency of migraine attacks was

**Table 1 Demographic data.** The patients' demographic data; no parameters were significantly differ between controls and migraine patients.

| Parameters | | Mean | SD | *p*-value* |
|---|---|---|---|---|
| Age (year) | Control | 36.45 | 9.273 | 0.942 |
| | Migraine | 36.29 | 9.449 | |
| VA OD (LogMAR) | Control | 0.150 | 0.2898 | 0.1340 |
| | Migraine | 0.065 | 0.1847 | |
| VA OS (LogMAR) | Control | 0.197 | 0.3217 | 0.0780 |
| | Migraine | 0.083 | 0.2201 | |
| BMI (kg/m$^2$) | Control | 22.480 | 3.1518 | 0.0550 |
| | Migraine | 24.060 | 3.8912 | |
| IOP OD (mmHg) | Control | 14.258 | 2.6726 | 0.2490 |
| | Migraine | 14.922 | 2.2517 | |
| IOP OS (mmHg) | Control | 14.042 | 2.4031 | 0.3898 |
| | Migraine | 14.881 | 3.5035 | |

**Note:**
* Independent t-test.

7.34 ± 7.65 times per month. The mean score of the Thai-MIDAS in migraine patients was 19.63 ± 20.44, and the majority were categorized as grade IV: severe disability (44.70%).

The mean thickness of the pRNFL, mRNFL, GCL, INL are displayed according to eye sides (left or right) in Table 2.

Data were evaluated firstly regarding the presence or absence of migraine. The mean thicknesses of the retinal layers from both eyes of the migraine group and controls are compared using an independent t-test as displayed in Table 3. Among the retinal layers studied in both groups, only one area, the superior-outer sector INL of migraine patients, was significantly thicker than controls, exhibiting 33.82 ± 2.46 μm *versus* 32.66 ± 2.34 μm, respectively ($p$ = 0.039, t = 2.10 and degree of freedom = 74). For ease of orientation of each area and layer of the retina, the mean thickness of each layer is illustrated using Garway-Heath and ETDRS sectors with controls in Fig. 1, migraine cases by eye in Fig. 2, and migraine cases by headache-side or non-headache-side eyes in Fig. 3.

Further analysis considering the side of the headache in relation to the eye side was done by ANOVA and *post-hoc* analysis, comparing control eyes, headache-side eyes, and non-headache-side eyes showed reduced thickness of the GCL at temporal-outer sector and mRNFL at the superior-outer sector of the headache-side eyes when compared to that of control (mean difference 2.14 ± 1.04, $p$ = 0.042, 95%CI [0.077–4.212]; mean difference 3.47 ± 1.64, $p$ = 0.038, 95%CI [0.205–6.742], respectively).

When only patients with unilateral headaches were included, and thicknesses of the nasal and temporal sides of each eye were grouped in respect to the optic pathway, no significant difference of retinal layers between headache-side eyes, non-headache-side eyes, and controls were found.

Correlation analysis in Table 4 showed a low positive correlation between the Thai-MIDAS score and frequency of attacks (r = 0.495, $p$ = 0.002, 95%CI [0.319–0.747]).

**Table 2  Thickness of each layer and area (sector) of the retina of controls and migraine patients.** The thickness of each sector of the retina from left and right eye of controls and migraine patients.

| Layer and sector by eye | | | Thickness ± SD (µm) | | | |
|---|---|---|---|---|---|---|
| | | | Control | | Migraine | |
| | | | Mean | Std. deviation | Mean | Std. deviation |
| pRNFL | Right eye | G | 104.50 | 18.10 | 107.24 | 7.80 |
| | | T | 79.71 | 12.36 | 78.24 | 14.98 |
| | | TS | 147.18 | 18.24 | 147.16 | 17.41 |
| | | NS | 114.37 | 16.52 | 119.95 | 21.15 |
| | | N | 77.97 | 14.36 | 78.82 | 15.63 |
| | | NI | 124.32 | 19.22 | 122.45 | 21.75 |
| | | TI | 155.87 | 17.80 | 154.13 | 20.25 |
| | Left eye | G | 106.05 | 10.09 | 106.58 | 7.71 |
| | | T | 79.13 | 12.42 | 77.47 | 14.49 |
| | | TS | 148.34 | 21.27 | 146.39 | 19.15 |
| | | NS | 125.26 | 18.67 | 133.00 | 22.50 |
| | | N | 69.68 | 11.71 | 73.16 | 13.68 |
| | | NI | 115.39 | 26.44 | 115.29 | 19.83 |
| | | TI | 159.42 | 15.17 | 155.66 | 18.77 |
| mRNFL | Right eye | Central | 13.74 | 14.17 | 10.92 | 2.10 |
| | | T-inner | 16.92 | 1.22 | 16.38 | 1.01 |
| | | S-inner | 24.74 | 4.52 | 23.11 | 3.20 |
| | | N-inner | 20.08 | 2.31 | 19.95 | 2.37 |
| | | I-inner | 24.05 | 3.18 | 23.95 | 2.84 |
| | | T-outer | 19.21 | 4.79 | 18.14 | 1.00 |
| | | S-outer | 40.55 | 7.08 | 37.57 | 3.97 |
| | | N-outer | 47.11 | 7.99 | 45.68 | 5.75 |
| | | I-outer | 40.00 | 4.81 | 38.81 | 5.24 |
| | Left eye | Central | 11.39 | 2.34 | 11.16 | 2.26 |
| | | T-inner | 16.29 | 0.93 | 16.26 | 1.20 |
| | | S-inner | 24.11 | 3.48 | 22.84 | 3.25 |
| | | N-inner | 19.84 | 2.16 | 19.37 | 1.70 |
| | | I-inner | 22.79 | 3.36 | 23.26 | 2.50 |
| | | T-outer | 17.92 | 1.24 | 18.74 | 4.68 |
| | | S-outer | 39.45 | 7.00 | 37.37 | 4.78 |
| | | N-outer | 46.53 | 6.14 | 43.55 | 7.29 |
| | | I-outer | 39.21 | 4.97 | 38.32 | 5.03 |
| GCL | Right eye | Central | 13.53 | 3.96 | 11.76 | 2.99 |
| | | T-inner | 45.84 | 5.25 | 46.22 | 4.72 |
| | | S-inner | 52.61 | 3.97 | 52.32 | 3.56 |
| | | N-inner | 51.39 | 4.51 | 50.73 | 3.81 |
| | | I-inner | 50.76 | 5.09 | 51.43 | 3.25 |
| | | T-outer | 37.84 | 4.04 | 36.76 | 3.72 |

(Continued)

| Layer and sector by eye | | | Thickness ± SD (µm) | | | |
|---|---|---|---|---|---|---|
| | | | **Control** | | **Migraine** | |
| | | | **Mean** | **Std. deviation** | **Mean** | **Std. deviation** |
| | | S-outer | 36.03 | 3.13 | 36.89 | 3.67 |
| | | N-outer | 40.45 | 3.42 | 40.49 | 3.45 |
| | | I-outer | 34.79 | 3.80 | 34.51 | 2.73 |
| | Left eye | Central | 13.34 | 4.04 | 12.37 | 4.04 |
| | | T-inner | 46.68 | 4.44 | 46.24 | 3.89 |
| | | S-inner | 52.13 | 3.93 | 52.45 | 3.57 |
| | | N-inner | 49.84 | 4.89 | 50.63 | 3.99 |
| | | I-inner | 50.55 | 5.34 | 50.61 | 4.40 |
| | | T-outer | 36.87 | 3.59 | 36.47 | 4.51 |
| | | S-outer | 35.68 | 3.14 | 36.18 | 3.40 |
| | | N-outer | 40.97 | 3.71 | 41.34 | 3.42 |
| | | I-outer | 35.39 | 3.69 | 35.05 | 3.61 |
| INL | Right eye | Central | 16.74 | 4.32 | 15.62 | 4.27 |
| | | T-inner | 37.11 | 3.29 | 36.78 | 3.87 |
| | | S-inner | 40.71 | 3.76 | 41.16 | 3.63 |
| | | N-inner | 40.29 | 3.59 | 40.32 | 3.38 |
| | | I-inner | 40.61 | 3.60 | 40.14 | 3.23 |
| | | T-outer | 34.42 | 2.53 | 34.24 | 2.65 |
| | | S-outer | 32.63 | 2.59 | 33.73 | 2.50 |
| | | N-outer | 35.08 | 2.46 | 35.16 | 2.39 |
| | | I-outer | 32.66 | 2.46 | 32.43 | 2.70 |
| | Left eye | Central | 16.39 | 4.27 | 15.71 | 4.79 |
| | | T-inner | 37.76 | 2.93 | 37.26 | 3.67 |
| | | S-inner | 41.66 | 3.68 | 40.79 | 3.56 |
| | | N-inner | 39.34 | 3.97 | 39.34 | 3.29 |
| | | I-inner | 40.39 | 4.04 | 40.47 | 3.58 |
| | | T-outer | 34.08 | 2.26 | 33.92 | 2.28 |
| | | S-outer | 32.68 | 2.34 | 33.92 | 2.61 |
| | | N-outer | 35.71 | 3.05 | 35.68 | 3.20 |
| | | I-outer | 33.16 | 2.97 | 32.74 | 2.71 |

**Note:**
G, global; T, temporal; TS, temporal-superior; NS, nasal-superior; N, nasal; NI, nasal-inferior; TI, temporal-inferior.

The severity scores also showed low positive correlations with the mRNFL thickness at the nasal-inner sector of the non-headache-side eyes (r = 0.448, $p$ = 0.048, 95%CI [−0.310 to 0.781]), and moderate negative correlations with the INL thickness at the superior-outer sector of the headache-side eye (r = −0.509, $p$ = 0.026, 95% CI [−0.770 to 0.014]).

**Table 3 Mean thicknesses of each retinal layers from both eyes of controls and migraine patients.** The thickness of each retinal sector from both eyes in controls and migraine patients.

| Layers and sectors both eyes | | Thickness ± SD (µm) | | | | df | t-Value | p-Value* |
|---|---|---|---|---|---|---|---|---|
| | | Control | | Migraine | | | | |
| | | Mean | Std. deviation | Mean | Std. deviation | | | |
| pRNFL | G | 105.28 | 12.16 | 106.91 | 7.62 | 56 | 0.70 | 0.486 |
| | T | 79.42 | 11.95 | 77.86 | 13.53 | 56 | 0.54 | 0.594 |
| | TS | 147.76 | 18.97 | 146.78 | 17.03 | 56 | 0.24 | 0.812 |
| | NS | 119.82 | 16.21 | 126.47 | 20.88 | 56 | 1.56 | 0.125 |
| | N | 73.83 | 11.66 | 75.99 | 13.84 | 56 | 0.74 | 0.465 |
| | NI | 119.86 | 20.63 | 118.87 | 19.21 | 56 | 0.22 | 0.830 |
| | TI | 157.64 | 15.59 | 154.89 | 18.32 | 56 | 0.71 | 0.483 |
| mRNFL | Central | 12.57 | 7.03 | 11.01 | 1.99 | 56 | 1.30 | 0.200 |
| | T-inner | 16.61 | 0.81 | 16.32 | 0.95 | 56 | 1.38 | 0.172 |
| | S-inner | 24.42 | 3.61 | 23.00 | 2.82 | 56 | 1.90 | 0.062 |
| | N-inner | 19.96 | 1.98 | 19.68 | 1.77 | 56 | 0.66 | 0.514 |
| | I-inner | 23.42 | 2.80 | 23.62 | 2.18 | 56 | 0.35 | 0.731 |
| | T-outer | 18.57 | 2.50 | 18.45 | 2.47 | 56 | 0.21 | 0.835 |
| | S-outer | 40.00 | 6.84 | 37.45 | 4.19 | 56 | 1.95 | 0.056 |
| | N-outer | 46.82 | 6.52 | 44.58 | 5.96 | 56 | 1.55 | 0.126 |
| | I-outer | 39.61 | 4.58 | 38.58 | 4.89 | 56 | 0.94 | 0.352 |
| GCL | Central | 13.43 | 3.77 | 12.03 | 3.28 | 56 | 1.73 | 0.089 |
| | T-inner | 46.26 | 4.59 | 46.22 | 3.87 | 56 | 0.05 | 0.962 |
| | S-inner | 52.37 | 3.83 | 52.34 | 3.39 | 56 | 0.04 | 0.971 |
| | N-inner | 50.62 | 4.50 | 50.66 | 3.46 | 56 | 0.05 | 0.963 |
| | I-inner | 50.66 | 4.91 | 50.96 | 3.29 | 56 | 0.31 | 0.756 |
| | T-outer | 37.36 | 3.47 | 36.66 | 3.88 | 56 | 0.82 | 0.417 |
| | S-outer | 35.86 | 2.99 | 36.57 | 3.46 | 56 | 0.96 | 0.342 |
| | N-outer | 40.58 | 3.37 | 39.79 | 3.20 | 56 | 0.85 | 0.401 |
| | I-outer | 35.09 | 3.58 | 34.81 | 2.97 | 56 | 0.37 | 0.712 |
| INL | Central | 16.65 | 4.07 | 16.05 | 4.67 | 56 | 0.50 | 0.620 |
| | T-inner | 37.43 | 2.84 | 37.01 | 3.56 | 56 | 0.57 | 0.573 |
| | S-inner | 41.18 | 3.38 | 40.91 | 3.33 | 56 | 0.36 | 0.720 |
| | N-inner | 39.82 | 3.56 | 39.73 | 2.80 | 56 | 0.12 | 0.908 |
| | I-inner | 40.50 | 3.42 | 40.26 | 3.08 | 56 | 0.32 | 0.747 |
| | T-outer | 34.25 | 2.28 | 34.11 | 2.37 | 56 | 0.26 | 0.792 |
| | S-outer | 32.66 | 2.34 | 33.82 | 2.46 | 56 | 2.11 | 0.039 |
| | N-outer | 35.39 | 2.63 | 35.43 | 2.73 | 56 | 0.06 | 0.952 |
| | I-outer | 32.91 | 2.57 | 32.61 | 2.57 | 56 | 0.51 | 0.615 |

Notes:
*$p < 0.05$.
df, degree of freedom; G, global; T, temporal; TS, temporal-superior; NS, nasal-superior; N, nasal; NI, nasal-inferior; TI, temporal-inferior.
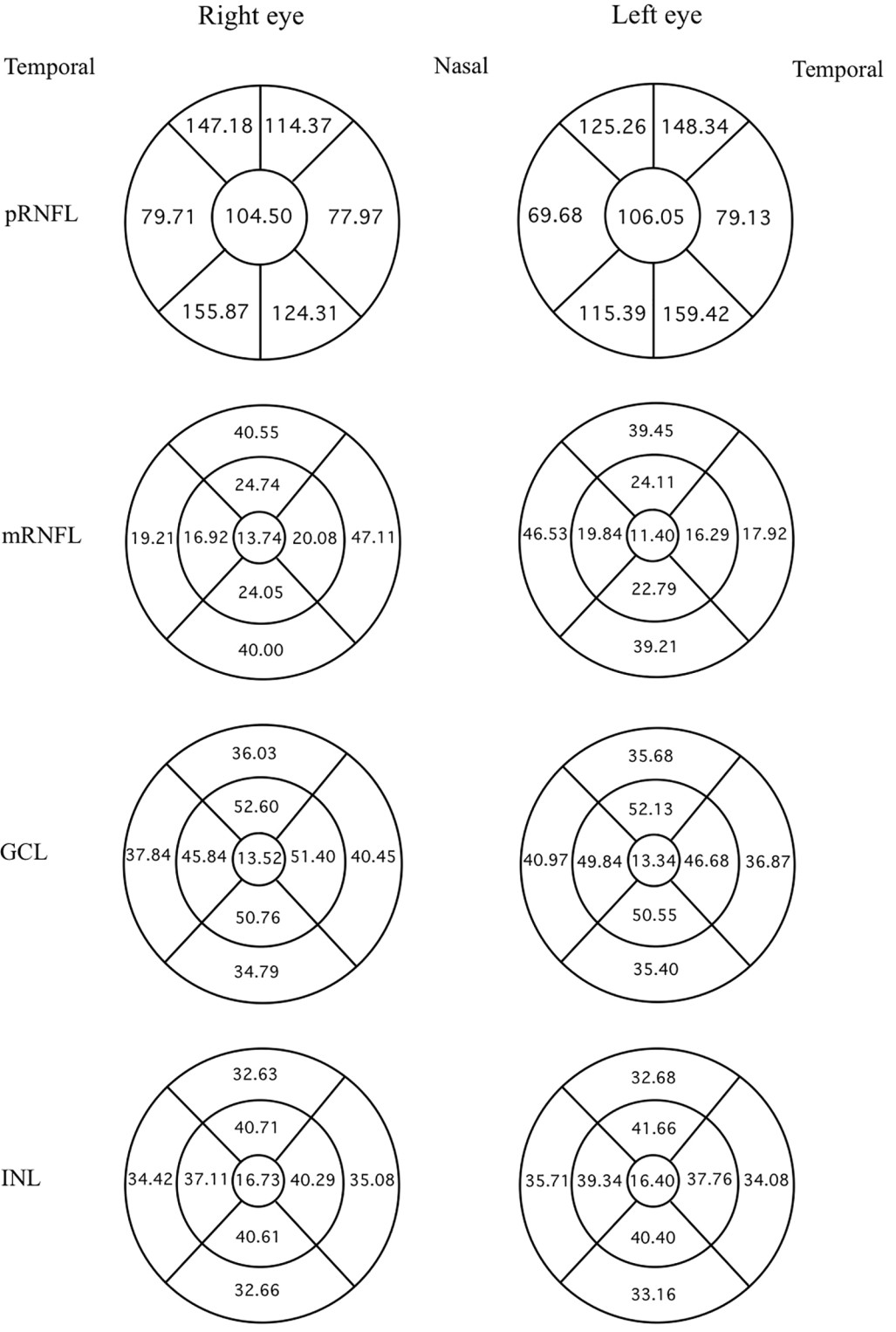

**Figure 1 Mean thicknesses of each sector of retinal layers by eye side in healthy controls.** Mean thicknesses of each sector of different retinal layers by eye side of healthy controls. First column displays the right eye, and second column the left. First row is the peripapillary retinal nerve fiber layer (pRNFL)

**Figure 1** (continued)
in Garway-Heath sector, inner circle shows the global average, the circumferential sectors from inner (nasal) are the inner, inner-superior, temporal-superior, temporal, temporal-inferior, and nasal-inferior sectors of the pRNFL, respectively, anti-clockwise for the right eye, clockwise for the left. From the second row, ETDRS map was used: the innermost circle shows the foveal thickness. The middle circumferential sectors, starting from the nasal side are the nasal-inner, superior-inner, temporal-inner, and inferior-inner sectors, respectively, anti-clockwise for the right eye and clockwise for the left. The outermost circumferential sectors, starting from the nasal side, are the nasal-outer, superior-outer, temporal-outer, and inferior-outer sectors of the macular retinal layers, respectively, anti-clockwise for the right eye, and clockwise for the left eye.                                                                                     

# DISCUSSION

Evaluation of mean RNFL (peripapillary and macular) thickness from both eyes of migraine patients in our study has found no significant reduction compared to controls. Extended investigation to other retinal structures, including GCL and INL, according to the hypothesis that these layers of the retina might constitute more appropriate morphological parameters for axonal damage than RNFL (*Yener & Korucu, 2019*), found significant thickening of the superior-outer sector of the INL in migraine patients compared to controls. Furthermore, we observed a significant thinning of the GCL at temporal-outer sector and mRNFL at the superior-outer sector of the headache-side eyes when compared to that of control eyes.

   The results of RNFL thickness in migraine patients of our study were contrary to previous studies (*Demircan et al., 2015*; *Acer et al., 2016*; *Gunes et al., 2018*; *Sirakaya et al., 2020*; *Ao et al., 2018*; *Abdellatif & Fouad, 2018*; *Gipponi et al., 2013*; *Martinez, Proupim & Sanchez, 2009*; *Yülek et al., 2015*; *Sorkhabi et al., 2013*; *Reggio et al., 2017*; *Tak, Sengul & Bilak, 2018*; *Feng et al., 2016*), systematic reviews, and meta-analyses, in which RNFL thickness was thinner in the migraine group than in the control group, most significantly in the superior and inferior quadrants (*Lin et al., 2021*). However, RNFL parameters may change more in migraine with aura than in migraine without aura (*El-Shazly et al., 2017*; *Ekinci et al., 2014*; *Ulusoy, Horasanli & Kal, 2019*). It has been speculated that when migraine attacks, changes of the optic nerve and RNFL blood supply may lead to hypoxic injury, which in turn leads to the death of retinal ganglion cells (*Ao et al., 2018*). Transneuronal retrograde degeneration (TRD), *i.e.*, the degeneration of a target neuron after the death of the presynaptic neuron or loss of presynaptic input, occurs in most neurons in both the infantile and adult central nervous systems. TRD has been observed after pathological damage to the central and peripheral nervous systems. There is also evidence of atrophy of ipsilateral retinal ganglion cells after occipital lobectomy in animals. *Park et al. (2013)* demonstrated that the RNFL was reduced in patients with cerebral infarction, providing evidence for TRD in retinal ganglion cells. Therefore, repeated onsets of migraine could result in the alteration of the structure and function of the occipital lobe (*Cowey, Alecxander & Stoerig, 2011*) as a subsequent change to retinal structures. *Yener & Korucu (2019)* and *Simsek, Aygun & Yildiz (2015)* reported no significant difference in RNFL measurements between patients and controls, which is similar to our study. These distinct results may be explained with the use of different methods and sample sizes, racial

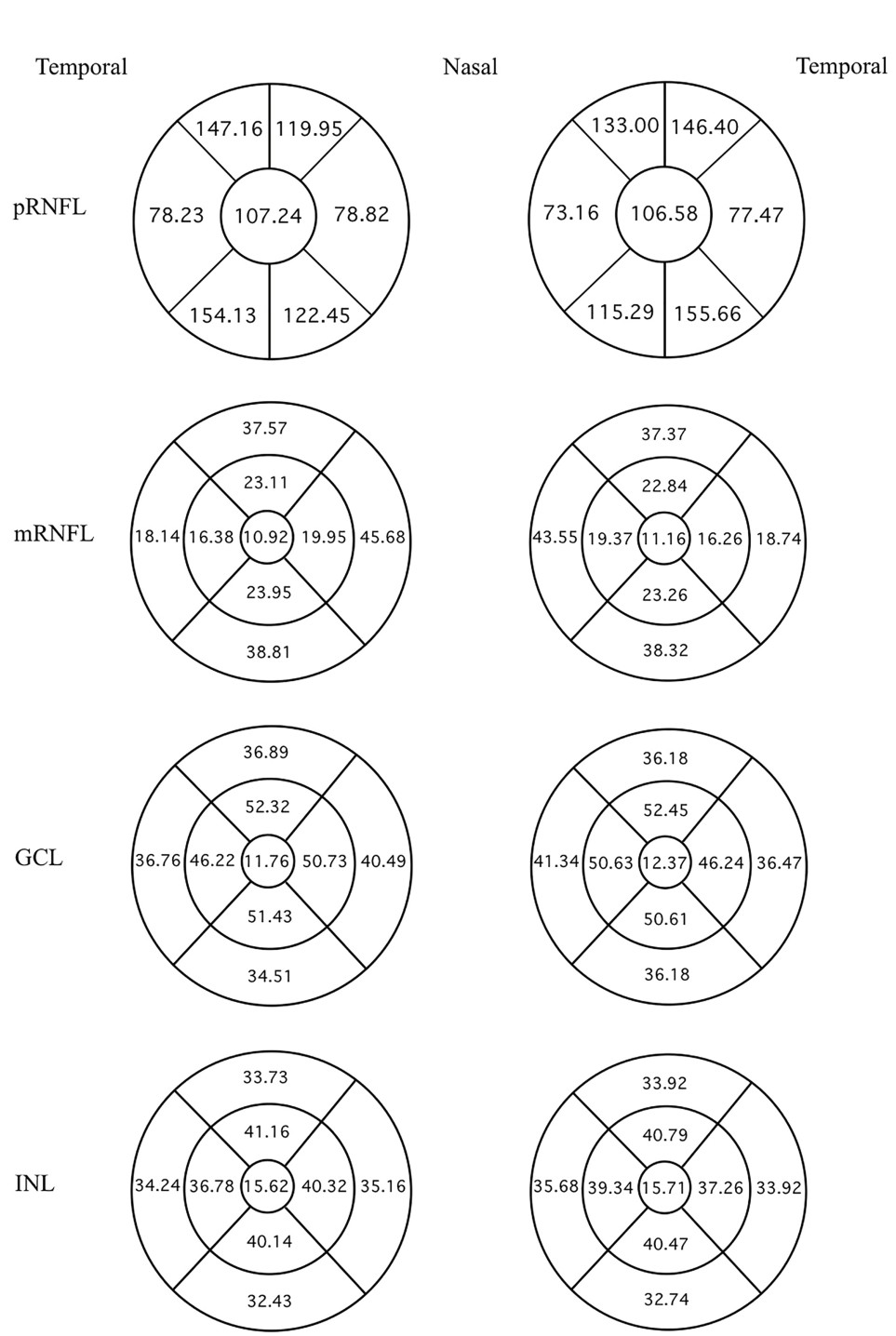

**Figure 2 Mean thicknesses of each sector of retinal layers by eye side in migraine patients.** Mean thicknesses of each sector of different retinal layers by eye side of migraine patients. The first column displays the right eye, the second column shows the left eye. Each row represents different location or layers measured.

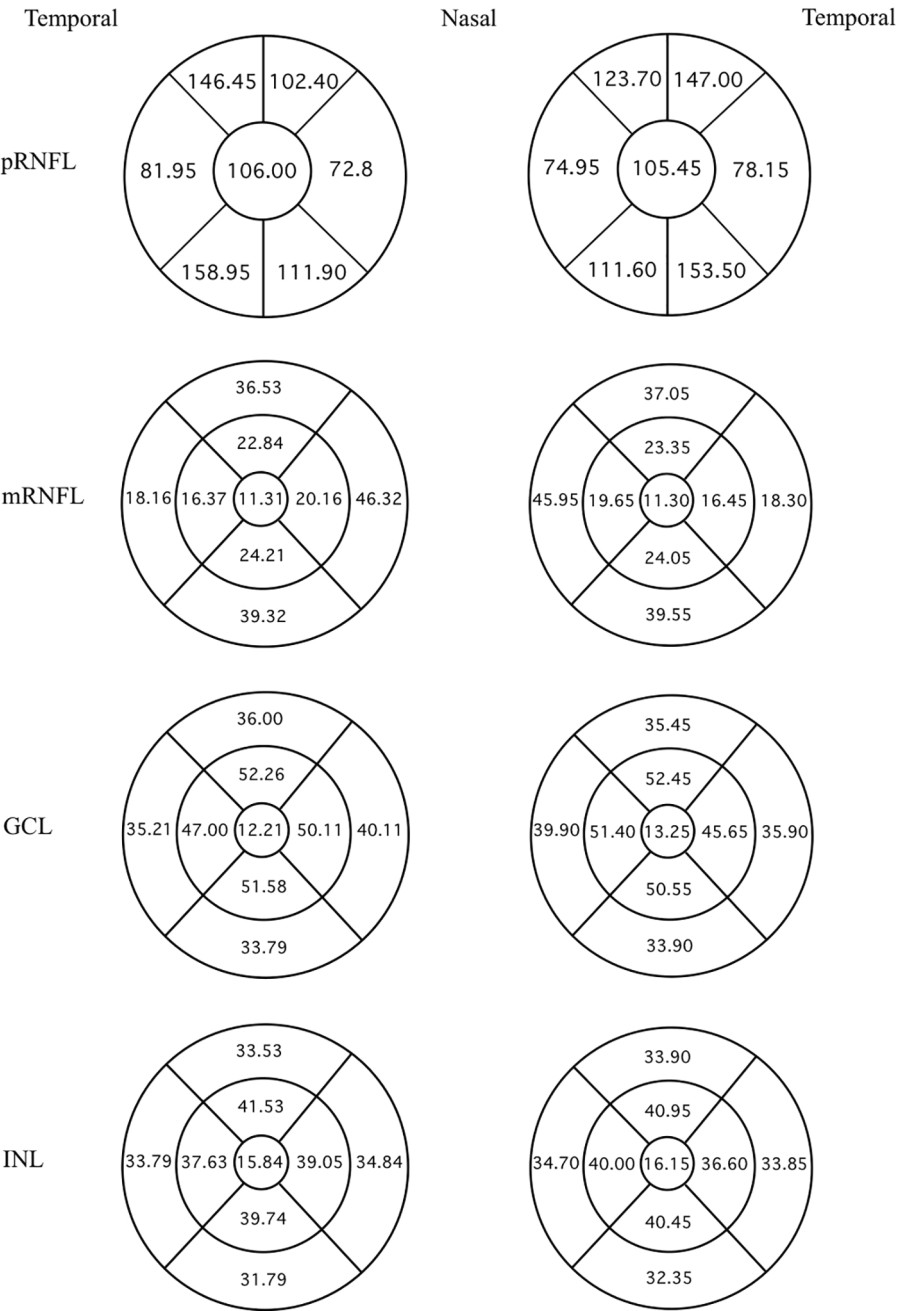

**Figure 3** **Mean thicknesses of each sector of retinal layers of the headache side eye and non-headache-side eye.** Mean thicknesses of each sector of different retinal layers of the headache side eye (1st column) and non-headache-side eye (2nd column). Please note that this figure represents the mean of the headache-side/non-headache-side eyes disregarding to the eye side. The headache-side eyes are shown in a diagram of the right eye, and non-headache-side eyes in a diagram of the left eye, this is for the readers' ease of orientation of the retinal sectors.

**Table 4 Correlation between Thai-MIDAS score and RNFL thickness.** The correlation between severity of migraine which evaluated by Thai-MIDAS score and the thickness of each retinal sector.

| Parameters | Thai-MIDAS score | | | | | | |
|---|---|---|---|---|---|---|---|
| | Pearson correlation (r) | p-value (2-tailed) | N | Bootstrap[c] | | | |
| | | | | Bias | Std. error | 95% Confidence interval | |
| | | | | | | Lower | Upper |
| Frequency (times/mo) | 0.495** | 0.002 | 38 | 0.028 | 0.115 | 0.316 | 0.763 |
| IpRNFLG | −0.155 | 0.515 | 20 | 0.008 | 0.158 | −0.431 | 0.175 |
| CpRNFLG | −0.049 | 0.837 | 20 | −0.015 | 0.168 | −0.438 | 0.260 |
| IpRNFLT | −0.053 | 0.825 | 20 | −0.036 | 0.253 | −0.516 | 0.466 |
| CpRNFLT | 0.177 | 0.455 | 20 | −0.020 | 0.237 | −0.317 | 0.608 |
| IpRNFLTS | −0.166 | 0.484 | 20 | 0.006 | 0.224 | −0.516 | 0.311 |
| CpRNFLTS | 0.204 | 0.388 | 20 | −0.046 | 0.245 | −0.501 | 0.513 |
| IpRNFLNS | −0.062 | 0.794 | 20 | 0.036 | 0.218 | −0.381 | 0.455 |
| CpRNFLNS | −0.099 | 0.679 | 20 | −0.047 | 0.239 | −0.672 | 0.242 |
| IpRNFLN | −0.047 | 0.843 | 20 | −0.008 | 0.219 | −0.460 | 0.357 |
| CpRNFLN | −0.023 | 0.924 | 20 | 0.002 | 0.215 | −0.410 | 0.395 |
| IpRNFLIN | −0.139 | 0.558 | 20 | 0.069 | 0.287 | −0.546 | 0.477 |
| CpRNFLNI | −0.302 | 0.195 | 20 | 0.033 | 0.223 | −0.613 | 0.243 |
| IpRNFLTI | −0.065 | 0.787 | 20 | −0.045 | 0.202 | −0.550 | 0.256 |
| CpRNFLTI | −0.070 | 0.768 | 20 | −0.022 | 0.155 | −0.473 | 0.180 |
| Innervating pRNFL | −0.167 | 0.481 | 20 | −0.011 | 0.159 | −0.516 | 0.140 |
| Non-innervating pRNFL | 0.027 | 0.909 | 20 | 0.009 | 0.162 | −0.256 | 0.383 |
| ImRNFLcen | 0.136 | 0.578 | 19 | 0.039 | 0.197 | −0.168 | 0.641 |
| CmRNFLcen | 0.223 | 0.360 | 19 | 0.025 | 0.167 | −0.093 | 0.575 |
| ImRNFLTinn | −0.074 | 0.762 | 19 | 0.011 | 0.183 | −0.423 | 0.325 |
| CmRNFLTinn | −0.295 | 0.221 | 19 | 0.034 | 0.236 | −0.618 | 0.312 |
| ImRNFLTSinn | 0.001 | 0.995 | 19 | −0.033 | 0.207 | −0.489 | 0.303 |
| CmRNFLTSinn | 0.245 | 0.313 | 19 | −0.068 | 0.292 | −0.428 | 0.627 |
| ImRNFLNinn | 0.156 | 0.522 | 19 | −0.007 | 0.184 | −0.276 | 0.503 |
| CmRNFLNinn | 0.448* | 0.048 | 20 | −0.072 | 0.307 | −0.305 | 0.780 |
| ImRNFLIinn | 0.259 | 0.284 | 19 | −0.015 | 0.216 | −0.208 | 0.658 |
| CmRNFLIinn | −0.415 | 0.069 | 20 | 0.086 | 0.315 | −0.769 | 0.381 |
| ImRNFLTout | −0.225 | 0.354 | 19 | −0.020 | 0.176 | −0.624 | 0.126 |
| CmRNFLTout | −0.361 | 0.129 | 19 | 0.018 | 0.190 | −0.659 | 0.102 |
| ImRNFLSout | −0.017 | 0.945 | 19 | −0.040 | 0.226 | −0.616 | 0.298 |
| CmRNFLSout | −0.040 | 0.872 | 19 | −0.048 | 0.247 | −0.660 | 0.320 |
| ImRNFLNout | 0.056 | 0.819 | 19 | −0.047 | 0.245 | −0.510 | 0.408 |
| CmRNFLNout | 0.164 | 0.502 | 19 | −0.097 | 0.356 | −0.609 | 0.604 |
| ImRNFLIout | −0.151 | 0.538 | 19 | −0.040 | 0.196 | −0.585 | 0.177 |
| CmRNFLIout | −0.195 | 0.424 | 19 | −0.054 | 0.212 | −0.660 | 0.156 |
| Innervating mRNFL | 0.102 | 0.679 | 19 | −0.066 | 0.298 | −0.567 | 0.488 |
| Non-innervating mRNFL | −0.072 | 0.771 | 19 | 0.035 | 0.206 | −0.382 | 0.415 |

| Parameters | Thai-MIDAS score | | | | | | |
|---|---|---|---|---|---|---|---|
| | Pearson correlation (r) | p-value (2-tailed) | N | Bootstrap[c] | | | |
| | | | | Bias | Std. error | 95% Confidence interval | |
| | | | | | | Lower | Upper |
| IGCLcen | 0.142 | 0.561 | 19 | 0.045 | 0.172 | −0.127 | 0.557 |
| CGCLcen | 0.446 | 0.056 | 19 | −0.022 | 0.206 | −0.022 | 0.792 |
| IGCLTinn | 0.275 | 0.255 | 19 | −0.046 | 0.260 | −0.304 | 0.687 |
| CGCLTinn | −0.122 | 0.617 | 19 | −0.035 | 0.201 | −0.633 | 0.201 |
| IGCLTSinn | 0.183 | 0.453 | 19 | −0.057 | 0.249 | −0.474 | 0.525 |
| CGCLTSinn | 0.163 | 0.506 | 19 | -0.088 | 0.328 | −0.670 | 0.595 |
| IGCLNinn | 0.136 | 0.578 | 19 | −0.035 | 0.211 | −0.340 | 0.443 |
| CGCLNinn | 0.338 | 0.157 | 19 | −0.096 | 0.339 | −0.464 | 0.745 |
| IGCLIinn | 0.216 | 0.374 | 19 | −0.113 | 0.367 | −0.603 | 0.685 |
| CGCLIinn | −0.220 | 0.365 | 19 | −0.028 | 0.190 | −0.646 | 0.132 |
| IGCLTout | 0.205 | 0.401 | 19 | −0.118 | 0.356 | −0.600 | 0.627 |
| CGCLTout | 0.041 | 0.866 | 19 | −0.062 | 0.279 | −0.537 | 0.478 |
| IGCLSout | −0.235 | 0.333 | 19 | −0.076 | 0.193 | −0.699 | 0.033 |
| CGCLSout | −0.300 | 0.212 | 19 | −0.051 | 0.137 | −0.641 | −0.107 |
| IGCLNout | −0.140 | 0.567 | 19 | −0.026 | 0.154 | −0.517 | 0.122 |
| CGCLNout | −0.197 | 0.420 | 19 | −0.015 | 0.166 | −0.561 | 0.140 |
| IGCLIout | −0.066 | 0.788 | 19 | −0.053 | 0.214 | −0.616 | 0.238 |
| CGCLIout | 0.346 | 0.147 | 19 | −0.151 | 0.412 | −0.653 | 0.774 |
| Innervating GCL | 0.099 | 0.686 | 19 | −0.081 | 0.316 | −0.620 | 0.549 |
| Non-innervating GCL | −0.040 | 0.870 | 19 | −0.067 | 0.270 | −0.639 | 0.365 |
| IINLcen | 0.019 | 0.940 | 19 | 0.045 | 0.242 | −0.376 | 0.570 |
| CINLcen | 0.167 | 0.494 | 19 | 0.021 | 0.159 | −0.096 | 0.525 |
| IINLTinn | −0.090 | 0.715 | 19 | −0.001 | 0.166 | −0.398 | 0.291 |
| CINLTinn | −0.204 | 0.402 | 19 | −0.004 | 0.173 | −0.504 | 0.183 |
| IINLTSinn | −0.310 | 0.197 | 19 | 0.002 | 0.199 | −0.625 | 0.204 |
| CINLTSinn | −0.036 | 0.883 | 19 | −0.001 | 0.189 | −0.410 | 0.350 |
| IINLNinn | −0.119 | 0.627 | 19 | 0.0058 | 0.245 | −0.413 | 0.534 |
| CINLNinn | −0.082 | 0.739 | 19 | 0.014 | 0.187 | −0.384 | 0.373 |
| IINLIinn | 0.037 | 0.880 | 19 | −0.004 | 0.169 | −0.306 | 0.361 |
| CINLIinn | −0.241 | 0.321 | 19 | 0.0013 | 0.230 | −0.634 | 0.257 |
| IINLTout | −0.170 | 0.486 | 19 | −0.035 | 0.169 | −0.577 | 0.101 |
| CINLTout | −0.040 | 0.870 | 19 | −0.010 | 0.166 | −0.436 | 0.280 |
| IINLSout | −0.509* | 0.026 | 19 | 0.030 | 0.197 | −0.746 | 0.067 |
| CINLSout | −0.398 | 0.091 | 19 | 0.022 | 0.215 | −0.742 | 0.136 |
| IINLNout | −0.192 | 0.430 | 19 | 0.013 | 0.222 | −0.580 | 0.330 |
| CINLNout | −0.136 | 0.579 | 19 | −0.015 | 0.157 | −0.459 | 0.161 |
| IINLIout | −0.200 | 0.411 | 19 | 0.005 | 0.201 | −0.572 | 0.250 |

(Continued)

| Parameters | Thai-MIDAS score | | | | | | |
|---|---|---|---|---|---|---|---|
| | Pearson correlation (r) | p-value (2-tailed) | N | Bootstrap[c] | | | |
| | | | | Bias | Std. error | 95% Confidence interval | |
| | | | | | | Lower | Upper |
| CINLIout | 0.047 | 0.849 | 19 | −0.044 | 0.246 | −0.543 | 0.418 |
| Innervating INL | −0.151 | 0.537 | 19 | −0.013 | 0.155 | −0.468 | 0.159 |
| Non-innervating INL | −0.193 | 0.429 | 19 | 0.005 | 0.216 | −0.546 | 0.352 |

Notes:
[*] Correlation is significant at the 0.05 level (2-tailed).
[**] Correlation is significant at the 0.01 level (2-tailed).
[c] Unless otherwise noted, bootstrap results are based on 1,000 bootstrap samples.
I, headache-side eye; C, non-headache-side eye; Cen, center; Inn, inner; Out, outer; G, global; T, temporal; TS, temporal-superior; NS, nasal-superior; N, nasal; NI, nasal-inferior; TI, temporal-inferior.

differences, and the variation of migraine parameters such as severity of headache, duration of headache, duration of attack, and frequency of attacks.

The extended investigation of INL layer in our results found significant thickening of the superior-outer sector in migraine patients compared to controls. The INL is composed of three neuronal cell classes (horizontal, bipolar, and amacrine cells) and one class of glia (Müller cells). Müller cell dysfunction in the fovea has been specifically linked to degenerative diseases affecting the macula (*Masri et al., 2021*). The INL is mostly studied in central demyelinating diseases, multiple sclerosis, and neuromyelitis optica (*Balk et al., 2019*; *Cellerino et al., 2019*; *Gelfand et al., 2013*). *Balk et al. (2019)* demonstrated the relationship between the thickening of the INL, physical disability, and disease activity in multiple sclerosis. The underlying mechanism responsible for the thickening of the INL remains unknown. The INL is embedded between the superficial vascular plexus and the deep capillary plexus, whereas both plexuses and Müller cells can absorb the interstitial fluid. One suggested mechanism involves inflammation-related dynamic fluid shifts (*Spaide, 2016*). Dynamic retinal layer volume changes can be explained by fluid shifts due to a combination of osmotic and hydrostatic gradients in the retinal lymphatic system (*Petzoid, 2016*). In the process of inflammation, diffusion of fluid from the retinal blood vessels increases, leading to an increase in INL thickness. Therefore, this finding supports that inflammation might be involved in the pathophysiology of migraine, leading to thickening of the INL. Moreover, Müller cell dysfunction can result from TRD. Another suggested mechanism is the pathology of Müller cells, which would impair the absorption of interstitial fluid, also resulting in increased INL thickness (*Lujan & Horton, 2013*).

Our study evaluated specifically in unilateral headache migraine, by observed the changed in retinal structural in headache-side eyes compared to contralateral-eyes and controls. There were significantly reduced in thickness of the GCL at temporal-outer sector and mRNFL at the superior-outer sector of the headache-side eyes. There was only one study which mention of lateralization of headache and RNFL thickness (*Gunes et al., 2016*). *Gunes et al. (2016)* investigated the relationship between migraine headache

lateralization and RNFL thickness in migraine patients with unilateral headache. They found the thinning of RNFL was higher on the side of the headache compared to the contralateral side, but the difference was not statistically significant. So, our study is the first study that found the significant relation of RNFL thinning on the headache-side eye. These results may explained by the reduction of blood flow and vasospasm during migraine episodes which usually occur in one hemisphere (*Killer, Forrer & Flammer, 2003*). There are some studies of unilateral involvement in migraine patients (*Friberg et al., 1991*; *Hougaard et al., 2015*). *Friberg et al. (1991)* found reduced blood flow velocities in the middle cerebral artery on the affected side during migraine episodes. *Hougaard et al. (2015)* identified thinning of the ipsilateral hemisphere cortical thickness in the inferior frontal gyrus. The finding of more prominent thinning in some part RNFL and GCL in headache-side eye might be related with lateralized structural abnormalities in the cortex, indicating that the hemisphere on the side of pain is affected more.

The positive correlation between the Thai-MIDAS score and frequency of attacks, although not surprising, affirms the nature of the disease. The correlation between the duration of migraine history, attack frequency, were not clear, like several studies that did not find any correlation similar to ours (*Simsek, Aygun & Yildiz, 2015*; *Labib et al., 2020*); nevertheless, inverse correlations have also been reported (*Acer et al., 2016*; *Martinez, Proupim & Sanchez, 2009*; *Reggio et al., 2017*; *Gunes et al., 2016*). Interestingly, our data showed a negative correlation between INL in the superior outer sector of the headache-side eyes despite being the only area showing significantly increased thickness when compared to controls. Although it seemed counterintuitive that the INL of the headache-side eyes negatively correlated with the disability scores, it may be explainable. In early or mild severity migraine, the pathogenesis could be explained by the transneuronal retrograde degeneration (TRD) that may cause cellular and interstitial inflammation and edema, most likely starting at the Müller cell, which was previously reported in multiple sclerosis and neuromyelitis optica, resulting in increased INL thickness (*Balk et al., 2019*; *Cellerino et al., 2019*; *Gelfand et al., 2013*; *Lujan & Horton, 2013*). On the contrary, in more severe migraine with higher disability scores, the TRD is more chronic and result in cellular loss and atrophy, resulting in thinner INL. However, these discrepant outcomes may also have been due to incidental variations in the studied population. The relationships between INL thickness and migraine headache variables should be studied further.

This study still possesses some limitations. A larger multi-center study should be beneficial to show stronger proof of eye-brain interaction of migraine.

## CONCLUSIONS

The thickness of the superior outer subfield of INL was significantly thicker in migraine patients than in healthy subjects. This finding may be related to the inflammation and Müller cell dysfunction that occurs in migraine. There was significant thinning of the GCL at temporal-outer sector and mRNFL at the superior-outer sector of the headache-side eyes which may related to headache lateralization, focal vasospasm and cortical changed. Correlation showed a positive correlation between the Thai-MIDAS score and frequency

of attacks, the nasal-inner mRNFL thickness, of the non-headache-side eyes, but negative correlations with the superior-outer INL thickness of the headache-side eyes.

### Funding

This work is supported by Walailak University Research grant number WUH-IRG-64-052. The funders had no role in study design, data collection and analysis, decision to publish, or preparation of the manuscript.

### Grant Disclosures

The following grant information was disclosed by the authors:
Walailak University Research: WUH-IRG-64-052.

### Competing Interests

The authors declare that they have no competing interests.

### Author Contributions

- Lunla Udomwech conceived and designed the experiments, performed the experiments, analyzed the data, prepared figures and/or tables, authored or reviewed drafts of the article, and approved the final draft.
- Rini Sulastiwaty analyzed the data, authored or reviewed drafts of the article, and approved the final draft.
- Doungkamol Siriarchawawat conceived and designed the experiments, performed the experiments, analyzed the data, prepared figures and/or tables, authored or reviewed drafts of the article, and approved the final draft.

### Human Ethics

The following information was supplied relating to ethical approvals (*i.e.*, approving body and any reference numbers):
Walailak Ethics Committee.

### Data Availability

The raw data is available in the Supplemental File.

### Supplemental Information

Supplemental information for this article can be found online at http://dx.doi.org/10.7717/peerj.17454#supplemental-information.

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
