# Peer review of "Migraine eye: correlation between migraine and the retina"

_PeerJ, doi:10.7717/peerj.17454_

## Round 0.1 · original submission · Major Revisions

Major revisions are required including statistical section please make sure all these are done when you resubmit.

·

Basic reporting

The authors need to provide sufficient statistical parameters other than the p-value, such as the t-value and degree of freedom. (Line 126,134,140,142)

Experimental design

I am concerned about the correlation analysis. It is unclear if the correlation with the thicknesses at a specific eye location was performed for all the locations independently (Bootstrap search approach ) or was performed only for nasal-inner mRNFL and superior-outer INL (hypothesis-driven approach). In the former case, the authors need to consider multiple comparisons for the statistics since the false positives would be high. In the latter case, the authors need to justify deriving these two hypotheses.

Validity of the findings

1. The authors so often make claims without showing statistical results. For example, (Line 136) it is claimed no significant difference in retinal layers between headache-side eyes, non-headache-side eyes, and controls was found. The authors should provide statistics for those claims.

2. The correlation analysis showed moderate negative correlations with the INL thickness at the superior-outer sector of the headache-side eye (Line 142), which suggest thinner INL was associated with more severe migraine. However, they also showed the superior-outer sector INL of migraine patients, was significantly thicker than controls. The authors need to explain the discrepancy.

Reviewer 2 ·

Basic reporting

Satisfactory

Experimental design

Satisfactory

Validity of the findings

Satisfactory

Additional comments

Minor Correction. Please refer to the attachment file.

Annotated reviews are not available for download in order to protect the identity of reviewers who chose to remain anonymous.

---

## Round 0.2 · Minor Revisions

Please complete this :- To follow previous comment #2 in this section, the authors need to clarify the procedure of Bootstrap analysis which aims to correct for multiple corrections in the correlation analysis.

·

Basic reporting

no comment

Experimental design

no comment

Validity of the findings

To follow previous comment #2 in this section, the authors need to clarify the procedure of Bootstrap analysis which aims to correct for multiple corrections in the correlation analysis.

Reviewer 2 ·

Basic reporting

Satisfactory

Experimental design

Satisfactory

Validity of the findings

Satisfactory

Additional comments

Corrections made are acceptable. Overall article is satisfactory.

---

## Round 0.3 · accepted · Accept

Thank you.

Your manuscript has been accepted.

·

Basic reporting

no comment

Experimental design

no comment

Validity of the findings

no comment